# Effects of Simulated Nitrogen Deposition and Micro-Environment on the Functional Traits of Two Rare and Endangered Fern Species in a Subtropical Forest

**DOI:** 10.3390/plants11233320

**Published:** 2022-12-01

**Authors:** Lingbo Ji, Liping Wei, Lingling Zhang, Yuanqiu Li, Yang Tian, Ke Liu, Hai Ren

**Affiliations:** 1CAS Engineering Laboratory for Vegetation Ecosystem Restoration on Islands and Coastal Zones, South China Botanical Garden, Chinese Academy of Sciences, Guangzhou 510650, China; 2University of Chinese Academy of Sciences, Beijing 100049, China; 3Key Laboratory of Vegetation Restoration and Management of Degraded Ecosystems, South China Botanical Garden, Chinese Academy of Sciences, Guangzhou 510650, China; 4Shimentai National Natural Reserve, Yingde 513000, China

**Keywords:** growth traits, defense traits, reproductive traits, canopy N addition

## Abstract

Although the effects of N deposition on forest plants have been widely reported, few studies have focused on rare and endangered fern species (REFs). Information is also lacking on the effects of micro-environments on REFs. We investigated the effects of N addition (canopy and understory N addition, CAN, and UAN) and micro-environments (soil and canopy conditions) on the functional traits (growth, defense, and reproduction; 19 traits in total) of two REFs—*Alsophila podophylla* and *Cibotium baromet*—in a subtropical forest in South China. We found that, compared to controls, CAN or UAN decreased the growth traits (e.g., plant height, H) of *C. baromet*, increased its defense traits (e.g., leaf organic acid concentrations, OA), delayed its reproductive event (all-spore release date), and prolonged its reproductive duration. In contrast, *A. podophylla* showed increased growth traits (e.g., H), decreased defense traits (e.g., OA), and advanced reproductive events (e.g., the all-spore emergence date) under CAN or UAN. Meanwhile, the negative effects on the *C. baromet* growth traits and *A. podophylla* defense traits were stronger for CAN than for UAN. In addition, the soil chemical properties always explained more of the variations in the growth and reproductive traits of the two REFs than the N addition. Our study indicates that, under simulated N deposition, *C. baromet* increases its investment in defense, whereas *A. podophylla* increases its investment in growth and reproduction; this may cause an increasing *A. podophylla* population and decreasing *C. baromet* population in subtropical forests. Our study also highlights the importance of considering micro-environments and the N-addition approach when predicting N deposition impact on subtropical forest REFs.

## 1. Introduction

Burning fossil fuels, producing fertilizers, and other human activities add substantial amounts of nitrogen (N) to the atmosphere, leading to an increase in N deposition worldwide [1,2]. This increase has major effects on the growth, defense, and reproduction of forest plant species [3,4,5,6]. However, previous studies on the response of forest plant species to N deposition mainly concern dominant species [7,8,9,10]. Knowledge is lacking related to the effects of N deposition on rare and endangered fern species (REFs). Studying REFs is important and urgent because they are highly sensitive to environmental change.

When studying the effects of environmental change, including N deposition, on forest plant species, trait-based approaches are increasingly being used as the traits reflect plant growth, and the defense and reproductive strategies [11]. For example, enhanced plant growth traits (e.g., height, specific leaf area, and leaf nutrient concentrations) often indicate suitable environments [12,13]. However, when plants are subjected to harsh environmental stress (e.g., pollution or herbivory), they can strengthen their defenses by producing more secondary metabolites such as lignin and phenolics [8,14]. Plants may also delay their reproductive phenology to adapt to environmental changes such as a decreasing nutrient supply [15,16]. However, studies to date have concentrated mainly on the responses of dominant species, with research gaps on the impact of the N deposition on the functional traits of REFs [3,5,17,18].

Furthermore, among growth, defense, and reproductive traits, previous studies including simulated N deposition have mainly focused on growth response (rather than defense or reproduction) in forest ecosystems [17,19,20], with inconsistent conclusions. For example, N addition directly enhanced leaf N and P concentrations, but declines or non-significant changes in leaf N and P concentrations were also found [21,22,23]. These contrasting results indicate that the effect of N deposition on forest plants is context dependent [23,24]. In fact, in addition to having direct effects, N deposition is likely to indirectly affect plant functional traits by modifying soil chemical properties and other micro-environmental conditions [10,25]. Some studies found that N addition can cause soil acidification, the leaching away of nutrients other than N, and reduce soil organic matter by inhibiting litter decomposition [4,26]. It is also possible that local variations in soil chemical properties or light availability can override the effects of N deposition on understory plants, which might explain the non-significant response of some plant traits to N addition [27]. We, therefore, assumed that the endangered fern species might also be sensitive to micro-environmental conditions influenced by N deposition.

Researchers have traditionally used understory or soil N additions to simulate forest N deposition [19,28]. As these methods of N addition bypass the interception of N by the forest canopy, they could exaggerate the effects of N deposition on the forest ecosystem and soil chemical properties by adding all N directly to the soil without any reduction by the canopy [9,29]. Therefore, in recent years, studies have begun to explore canopy N addition and its effects on the community structure of soil fauna and flora [15,30,31,32], water use efficiency and xylem formation of dominant canopy trees species [18,33,34,35,36], and leaf nutrient and secondary metabolite concentrations of dominant understory species [5,17,21,37]. Among those studies, canopy N addition can induce increased survival stresses for some understory plants [5,17]; it can also promote the diversity of soil biota and macrofauna but shows no impact on plant diversity [15,31]. Canopy N addition promoted or showed no impact on the growth of dominant canopy tree species, depending on the specific tree species [18,38]. Moreover, compared with CAN, UAN always showed greater effects on the dominant understory species but weaker effects on the dominant tree species and soil fauna [17,18,38,39]. However, the effects of N addition on REFs and whether these ferns have similar responses to dominant species are still unknown.

The objective of this study was to investigate the impacts of simulated N deposition on the growth, defense, and reproductive traits (19 traits in total) of two REFs (*Cibotium barometz* and *Alsophila podophylla*). The two REFs are both fern species associated with subtropical areas [40]; *A. podophylla* has a funnel-shaped crown and aerial roots, whereas *C. barometz* does not. We also compared the ability of N addition and micro-environmental conditions (canopy cover and soil properties) to explain the variation in the REFs’ functional traits. Our experiment was conducted in an evergreen broad-leaved forest in subtropical South China, where the CAN (canopy N addition) and UAN (understory N addition) were applied.

We attempted to test the following hypotheses:

(1) N addition would decrease the growth traits of both REFs and increase their defense traits due to the environmental stress induced by N addition. For reproductive traits, N addition would delay reproductive events and prolong reproductive duration [3,41]. Meanwhile, under the same N-addition rate, the effect on traits would be greater for UAN than for CAN.

(2) N addition would directly increase soil nitrogen contents but reduce soil pH and other soil nutrients due to N’s potential effect on soil acidification and soil nutrient leaching or imbalance. Accordingly, micro-environmental conditions would also have significant impact on the functional traits of both REFs [17,42,43].

## 2. Results

### 2.1. Effects of N Addition on Traits

For the growth traits of *C. barometz*, H, LCC, and Nitrate were significantly lower under CAN50, CAN25, and CAN50 than in the control (0 kg N ha^−1^ yr^−1^), respectively (Figure 1 and Table 1). For the defense traits, relative to the control, Cellulose was significantly higher under UAN25 and CAN50, and OA was significantly higher under UAN50 and CAN50, but SP was significantly lower under CAN25, CAN50, and UAN50. For *C. barometz’s* reproductive traits, relative to the control, ASR occurred significantly later under UAN25 and CAN50, and RD was significantly longer under UAN25. Within each N-addition rate, the effects of CAN on *C. barometz’s* growth traits were mostly greater than those of UAN. Specifically, the effects of CAN50 on H and ASR were greater than those of UAN50, and the effect of CAN25 on LCC was greater than that of UAN25. The effects of UAN25 on Cellulose, ASR, and RD were greater than those of CAN25, but the effect of CAN50 on ASR was greater than that of UAN50.

For *A. podophylla*, three growth traits—H, LCC, and Nitrate—were significantly higher under UAN50, UAN50, and CAN25, respectively, than in the control (Figure 1 and Table 1). For the defense traits, Lignin and OA were significantly lower under CAN25 than in the control. For *A. podophylla* reproductive traits, ASE and ASM occurred significantly earlier under UAN50 than in the control. Within each N-addition rate, the effects of CAN on *A. podophylla* traits were often greater than those of UAN. Specifically, the effects of CAN25 on Nitrate, Lignin, and OA were greater than those of UAN25. There was one exception: the effect of UAN50 on H was greater than that of CAN50.

### 2.2. Micro-Environmental Variation and Its Effects on REF Traits

The micro-environmental factors (soil chemical properties and subcanopy cover) did not significantly differ among N treatments or with the control (Appendix A).

Among the six growth traits, five (H, SLA, LNC, LPC, and Nitrate) for *C. barometz*, and four (SLA, LCC, LNC, and LPC) for *A. podophylla* were better explained using the single-variable micro-environment models (especially soil AP) than by N treatment (Table 2). For the defense traits, LDMC and Lignin for *C. barometz*, and LDMC, Cellulose, SP, and TNC for *A. podophylla* were best explained using the micro-environment models (especially soil AP). Moreover, almost all of the reproductive traits of the two REFs (except for *C. barometz* ASR and RD, and *A. podophylla* FSM) were best explained using the micro-environment models (especially soil AK). Finally, no trait variation was adequately explained using the interaction models of Soil × N treatment or subcanopy cover × N treatment.

Concerning the effects of micro-environmental factors (Figure 2 and Appendix A), for soil N effects (N, NO_3_^−^-N, and NH_4_^+^-N), we found that higher soil NO_3_^−^-N can be related to delayed FSE for *C. barometz*. Lower soil pH could advance FSM for *C. barometz* and increased LCC for *A. podophylla*. For the other soil nutrient effects (C, P, AP, K, and AK), *C. barometz* LPC decreased with decreasing soil P. The decreased *A. podophylla* SLA, but its increased TNC, LDMC, and SP were found to increase with soil AP. Decreasing soil K showed the tendency to delay *C. barometz* ASE, and decreasing soil AK was related to delayed *A. podophylla* FSR. However, *A. podophylla* Cellulose increased with decreasing soil C. The subcanopy cover showed no effects on the traits of either REF.

## 3. Discussion

### 3.1. Effects of N Addition on REF Traits

The two REFs we studied are fern species widely distributed in a subtropical region [40]. Yet, there were some differences in their morphological and physiological structure. For example, *A. podophylla* has a funnel-shaped crown and aerial roots, whereas *C. barometz* does not. As we assumed, in general, we found decreased growth traits, increased defense traits, and delayed or prolonged reproductive traits for *C. baromet* under N addition. Specifically, N addition significantly decreased three growth traits—plant height, leaf carbon, and nitrate concentrations (H, LCC, and Nitrate). We, therefore, confirmed that excess N can inhibit the growth of *C. barometz*, which was consistent with previous studies on the same traits for dominant species [6,44] in subtropical forests. Moreover, the negative responses of the above three traits were only found under the canopy N addition—either CAN25 or CAN50. We, therefore, infer that, in addition to the direct negative effects of excess N, the indirect effect of canopy N addition can be more important in explaining growth traits. For example, previous studies found that canopy N addition can increase the absorption of soil nutrients and water by dominant trees [18,36,45], which probably induces stronger competition with understory plants, thus reducing the nutrient supply and inhibiting the height growth or nutrient accumulation of *C. barometz*. Concerning the *C. barometz* defense traits, two traits—leaf organic and cellulose concentrations (OA and Cellulose)—increased with canopy or understory N treatment. This indicates that the applied N solution created a series of stresses (e.g., oxidative stress) on the *C. barometz* leaves [7,14,37]. Yet, *C. barometz* SP, another defense trait, significantly decreased under N addition, possibly associated with a greater allocation of C to OA (a low-cost compound), which could decrease plant defense costs when C supplies are low [17,46,47]. Concerning *C. barometz* reproductive traits, N-addition treatment (CAN50 or UAN25) significantly delayed the all-spore release date (ASR) and significantly prolonged the reproductive duration (RD), which was consistent with previous findings that plants tend to allocate more resources to organs (e.g., roots) for resource acquisition under N addition, thus reducing the resources allocated to reproductive organs and inhibiting their development [3,48].

Contrary to our hypothesis, and unlike *C. barometz*, *A. podophylla* showed increased growth traits, decreased defense traits, and advanced reproductive events under N addition. The direct addition of large amounts of N (UAN50) to the understory layer increased the H and LCC of *A. podophylla*. This suggests that the levels of N in the soil might have been below the optimum for *A. podophylla,* contrary to our assumption [13,49]. This result is similar to some previous studies on dominant understory species (e.g., *Blastus cochinchinensis*) in subtropical forests, where increased LCC and photosynthesis (leaf maximum photosynthetic rate) were found [37,39,50,51] and explained by the increase in the distribution of nitrogen to the photosynthetic apparatus (e.g., Rubisco and chlorophyll). For *A. podophylla* defense traits, the reduction in leaf lignin concentration (Lignin) and OA under CAN25 can be explained by the theory that N enrichment decreases the production of the C-based secondary compounds that are important for defense, indicating the alleviation of environmental stress [8,43,52]. For *A. podophylla* reproductive traits, the all-spore emergence date and maturation date (ASE and ASR) were advanced by UAN50, which might indicate increasing reproductive efficiency due to N addition [3,42,48]. This was consistent with some previous studies on grasslands, which showed that the increased resource availability by N addition can make plants allocate more resources (e.g., nutrients) to their reproductive organs, thus accelerating the development of reproductive organs and advancing the reproductive events [3,41,53,54]. Advancing reproductive events may help *A. podophylla* avoid competition for resources (e.g., soil nutrients) with other species by further staggering their reproductive period [54,55,56]. The contrary responses for the traits of the two studied species suggest that, although the two REFs are both ferns typical of the subtropical area, their environmental demands and adaptation strategies are not necessarily the same. Furthermore, the funnel-shaped crown and aerial roots were observed for *A. podophylla*, which were lacking for *C. barometz*. Those differences would reflect the difference in the ability of the two REFs to absorb or compete for natural resources (e.g., water and nutrients), hence resulting in their contrary responses to N addition [57,58,59]. Some studies on grassland also found that plant species with distinct resource acquisition ability (e.g., grass vs. rare forbs) showed contrary responses to N-addition treatment [41,53,54].

Finally, though the two N-addition approaches showed different effects on the traits of the two REFs, CAN generally displayed greater effects than UAN under the same N-addition rate. This was contrary to our hypothesis and the common previous findings on dominant understory species [5,17,37]. This indicates that the role of the canopy tree layer in nutrient cycling may be more complicated than we supposed; canopy trees may not only intercept N but may also cycle N from the canopy to the roots, thus influencing the whole forest ecosystem and, ultimately, affecting REFs. Previous studies have reported the effects of canopy openness or canopy tree species on understory plant functional traits. For instance, Jiang et al. [60] found a positive correlation between canopy openness and the leaf phosphorus concentrations of understory plants; Wang et al. [61] showed that canopy tree species *Cunninghamia lanceolata* decreased the diameters of the first- and second-order fine roots of the understory species *Loropetalum chinensis* through the integrated effects of nutrient competition and litter input. Further studies are needed to explain this ecosystem process.

### 3.2. Micro-Environmental Variation and Its Effects on REF Traits

Contrary to our hypothesis, N addition did not result in any significant changes in the micro-environment factors at our experimental site after the 8-year N-addition treatment. This can be explained by a very recent study at our experimental plots by Tian et al. [62]. They found that plants, especially tree species, absorbed and stored substantial additional N from soil through their roots (increasing the plant N pool by 120–412%), so that the soil N did not show significant variation under N-addition treatment. The insignificant change in the soil properties in response to N addition could also be found in other forest ecosystems. For example, a study by Yang et al. [63] in temperate forests showed that the nine-year N addition (100 kg N ha^−1^ yr^−1^) did not cause changes to the soil N and pH, and they also attribute this to the plants’ absorption of additional N. In addition, some studies (in subtropical or temperate forests) showed that only under very high N-addition rates (100 and 150 kg N ha^−1^ yr^−1^ in their studies vs. 50 kg N ha^−1^ yr^−1^ in ours) can soil pH and N be affected [4,10,23]. However, micro-environmental factors (especially soil AP) often appeared to be more important than N treatment in explaining the growth and reproductive traits of both REFs and the defense traits of *A. podophylla*. Therefore, the micro-environment needs to be considered in order to improve REF growth and reproduction in subtropical forests.

Consistent with our hypothesis, higher soil NO_3_^−^-N delayed *C. barometz* FSE, which was also found in other forests [64,65,66]. The effects of the other soil nutrients (P, AP, K, and AK, excluding soil N) on the REF traits support our hypothesis that their low contents would correlate to reduced growth traits, delayed reproductive events, and prolonged reproductive periods for both REFs, and to higher defense trait values for *A. podophylla*. For example, the growth trait SLA decreased and the defense traits (LDMC, TNC, SP) increased for *A. podophylla* with decreasing soil AP [8,67,68]. However, lower soil pH could advance *C. barometz* FSM and increase *A. podophylla* LCC, possibly because both REFs prefer more acidic soil than that indicated by a pH value in the normal range [40,69]. Meanwhile, there was no effect of subcanopy cover on any traits for either REF, indicating that variations in light availability were not strong enough to override the effects of N addition or soil properties.

## 4. Materials and Methods

### 4.1. Study Site

Our research was carried out in an evergreen broadleaved forest (24°22′–24°31′ N, 113°05′–113°31′ E) at the Shimentai National Nature Reserve, Guangdong Province, China (Figure 3). The region has a subtropical monsoon climate with alternating wet (April–September) and dry (November–March) seasons. The mean annual precipitation and temperature are 2364 mm and 20.8 °C, respectively [70]. The dominant tree species are *Castanea henryi* (Skan) Rehd. et Wils., *Castanopsis eyrei* (Champ. ex Benth.) Tutch., and *Schima superba* Gardn. et Champ.; dominant shrubs include *Blastus cochinchinensis* Lour. *Psychotria rubra* Wall. and *Ardisia quinquegona* Blume; and the dominant herbaceous species are *Alpinia chinensis* Hayata, *Hypolytrum latifolium* L. C. Rich. and *Sarcandra glabra* (Thunb.) Nakai. The background N deposition in the Shimentai National Nature Reserve was 34.1 kg N ha^−1^ yr^−1^ [70], which is twice the maximum empirical critical loads (17 kg N ha^−1^ yr^−1^) for N deposition on territorial ecosystems [71,72].

### 4.2. Rare and Endangered Fern Species (REFs)

The two REFs—*Cibotium barometz* (L.) J. Sm. and *Alsophila podophylla* Hook.–are included in Appendix II of the Convention on International Trade in Endangered Species of Wild Fauna and Flora and are listed as Grade II in National Key Protected Wild Plants [73,74]. Both REFs have high ornamental and medicinal values and are widely distributed in East and Southeast Asia (Appendix A) [40]. Their trunks are important and unique substrates for epiphytes (e.g., orchids, ferns, and bryophytes), and their crowns can intercept litter, thus regulating the forest nutrient cycle [58,75].

### 4.3. Experimental Design

Our experiment had a randomized block design with four blocks (Figure 3), each of which contained five circular plots with a radius of 17 m. Four blocks were established at each forest site, and each treatment was replicated once within each of the four blocks. Within each block, the five treated plots were randomly assigned. In each block, each plot was treated with one of the following treatments: canopy N addition at 25 or 50 kg N ha^−1^ yr^−1^ (CAN25 or CAN50), understory N addition at 25 or 50 kg N ha^−1^ yr^−1^ (UAN25 or UAN50), and a control. The UAN was also used as a reference for the CAN effect. From 2013 to 2021, the CAN and UAN plots were sprayed with an NH_4_NO_3_ solution once a month from April to October, as described, in detail, by Zhang et al. [70]. In brief, the CAN was applied via a 35 m high tower (5–8 m above the canopy, with 4 sprinklers) built in the center of each CAN plot, whereas the UAN was applied using five sprinklers that were evenly distributed 1.5 m above the ground in each UAN plot. These sprinklers could turn 180–360° and spray the N solution as far as 17 m, depending upon the pressure used. The targeted N solution was made by weighing appropriate amount of ammonium nitrate (NH_4_NO_3_) and mixing with surface water drained from a nearby pond; it was transported to sprinklers through various polyvinylchloride pipes. The control plots were left untreated. As the total quantity of solution applied per year accounted for <1% of the total annual precipitation of the forest site, any confounding effect of additional water was negligible.

### 4.4. Assessment of Plant and Soil Chemical Properties

In January 2021, one 100 m^2^ quadrat was designated in each plot in the four blocks. After surveying the vegetation in each quadrat and classifying it into two strata (canopy tree layer with DBH > 10 cm and subcanopy tree layer with DBH < 10 cm and height > 2 m), we estimated the percentage of cover of each individual tree in order to calculate the percentage of canopy cover and subcanopy cover in each quadrat. Soil samples (5 cm in diameter × 10 cm in depth) were collected from four random points in each quadrat and were mixed to yield one composite soil sample per quadrat [76]. For each sample, we measured soil pH (pH), soil organic carbon content (C), total nitrogen (N), total phosphorus (P), total potassium (K), nitrate-N (NO_3_^−^-N), ammonia-N (NH_4_^+^-N), available phosphorus (AP), and available potassium (AK). pH was determined using a pH meter at a water/soil ratio of 2.5:1.0. C, N, P, and AP contents were determined according to the potassium dichromate, Kjeldahl, molybdenum anti-colorimetric, and molybdenum blue methods. K and AK contents were determined using the flame photometry method and NO_3_^−^-N and NH_4_^+^-N contents using the ultraviolet spectrophotometry method [77,78,79].

### 4.5. Assessment of Functional Traits

We selected 19 traits in total to represent plant growth, defense, and reproduction. For the growth traits, we measured the height (H) of all individuals of the two REFs in each quadrat. We then selected 3–5 mature individuals per species per quadrat and collected 15 leaves from each individual. There are on average 4–5 mature individuals of *C. baromet* or *A. podophylla* in each 10 m × 10 m quadrat. The two REFs often have large crowns of which the diameter can reach about 2 m. The same leave samples were used to analyze growth traits and defense traits. For growth traits, specific leaf area (SLA) was the leaf area (measured using an LI-3000C leaf area meter; Li-COR, Lincoln, NE, USA) divided by the dry weight. Concentrations of leaf carbon (LCC) and nitrogen (LNC) were determined using the potassium dichromate method and the modified Kjeldahl method, respectively [77]. Concentrations of leaf phosphorus (LPC) and nitrates (Nitrate) were determined using the molybdenum anti-colorimetric method, on the one hand and the ultraviolet spectrophotometry, on the other [46,77].

For defense traits, leaf dry matter content (LDMC) was calculated by dividing leaf fresh weight by leaf dry weight (measured after 65 °C at 72 h). Concentrations of leaf soluble phenolics (SP) were determined using the anthrone reagents method [46]. Concentrations of leaf cellulose (Cellulose) and lignin (Lignin) were determined using acid detergent lignin methods [25]. Concentrations of leaf total non-structural carbohydrates (TNC) were the sum of the concentrations of soluble sugars and insoluble sugars, which were determined using the Folin–Ciocalteu method [17]. Quantities of leaf ash were determined acidimetrically after the leaf powder was combusted in a muffle furnace at 550 °C for 6 h, and concentrations of leaf organic acid (OA) were calculated by multiplying the difference between leaf ash and Nitrate concentrations by 62.1 [46].

Regarding reproductive traits, for each REF, at least three individuals with similar heights and diameters from each quadrat were marked in the study site, and events in their phenological development were monitored weekly from March to December in 2021. The spores of the two REFs appeared when the frond leaves were expanding; this moment was defined as “spore emergence”. When the sporangia turned dark, this was defined as “spore maturation”, and the appearance of almost empty sporangia was defined as “spore release” [80]. In total, we recorded six key phenological events and transformed these into Julian days [56,81]: (1) first spore emergence, maturation, or release date (FSE, FSM, and FSR); and (2) all-spore emergence, maturation, or release date (ASE, ASM, or ASR). The reproductive duration (RD) was defined as the duration from the first emergence of spores to the release of all spores and was calculated by subtracting FSR from ASR. Background information for all of the measured traits is shown in Table 3.

### 4.6. Statistical Analyses

Analyses of variance (ANOVAs) were used to compare the effects of the N-addition treatments (CAN25, CAN50, UAN25, and UAN50) on each trait and on two micro-environmental variables (soil chemical properties and subcanopy cover; canopy cover was not considered because it was close to 100% in all plots). Differences in the above indices between the N-addition treatments were determined using Tukey HSD post hoc tests. We performed Pearson correlation analysis to have a general view of the correlations between traits and soil chemical properties (see Appendix A and results descriptions in Appendix A). Next, we used general linear models to model the relationships between each trait (Table 3), and the variables are listed in Table 4, including N treatment, best soil variables (selected by the model comparison of soil chemical properties based on AICc), subcanopy cover, and their combined effect. We, therefore, applied two sets of parallel explanatory models to each trait: (i) single-variable models of the best soil variable (Soil), subcanopy cover, and N treatment; and (ii) interaction models of Soil/subcanopy cover and N treatment. For each trait of the two REFs, we used Akaike’s information criterion (AIC) [87,88] to assess the relative importance of models related to N-addition treatment, micro-environment, and their interaction effects. The AIC-based model comparison method was commonly used to estimate the relative quality of the statistical models [89,90]. The smaller the AIC value, the better the model fit. We ranked the models by their AICc (small-sample-corrected Akaike information criterion) and calculated the delta AICc value—the difference in AIC value between the best model and the model being compared. The model with the smallest AICc was selected as the final model that best explained trait variations, and the effects of the best micro-environmental models on traits were tested. All analyses were conducted using R version 3.5.2.

## 5. Conclusions

For the two rare and endangered ferns species we studied, *C. barometz* increased its investment in defense and decreased its investment in growth and reproduction under N addition. In contrast, *A. podophylla* showed enhanced growth and reproduction and a decreased investment in defense. These results imply that the population of *A. podophylla* may increase and that of *C. barometz* face the possibility of declining under high N deposition in subtropical forests. As they may have different responses to global change, *A. podophylla*, *C. barometz*, and other rare and endangered plants in subtropical forests may require different protection strategies. In addition, our study shows the importance of using a canopy N-addition method rather than the commonly applied understory N-addition method, as the latter can underestimate the negative impact of N deposition on some traits of the two rare and endangered ferns. Finally, our study also highlights the importance of considering the micro-environment when predicting the effects of N deposition on rare and endangered ferns, as its influence sometimes overrides the effects of N addition.

## Figures and Tables

**Figure 1 plants-11-03320-f001:**
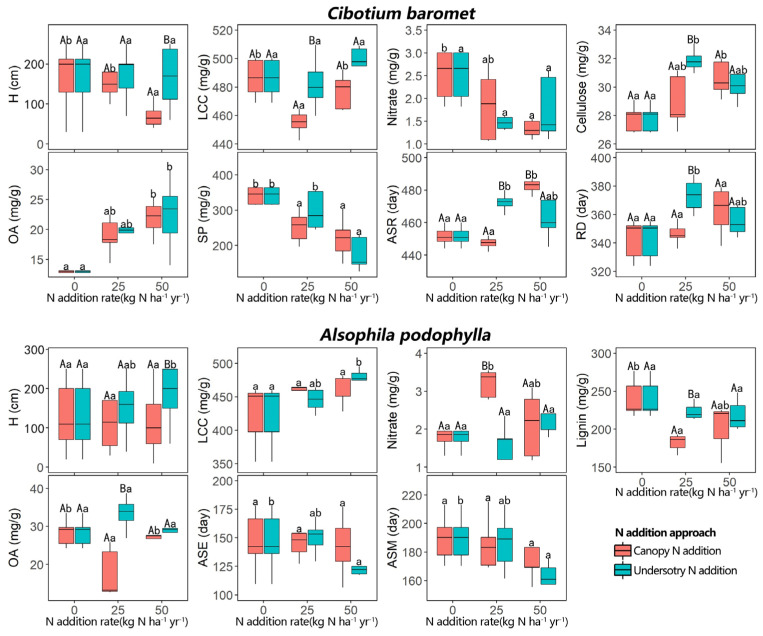
Effects of N-addition rate and approach on the functional traits of the two REFs. Only significant effects (Tukey HSD post hoc test, *p* < 0.05) were plotted. The different lower case and capital letters, respectively, indicate significant differences among N-addition rates within each N-addition approach and differences between N-addition approaches within each N-addition rate. Abbreviations of the traits and ecological variables, respectively, are explained in Tables 3 and 4.

**Figure 2 plants-11-03320-f002:**
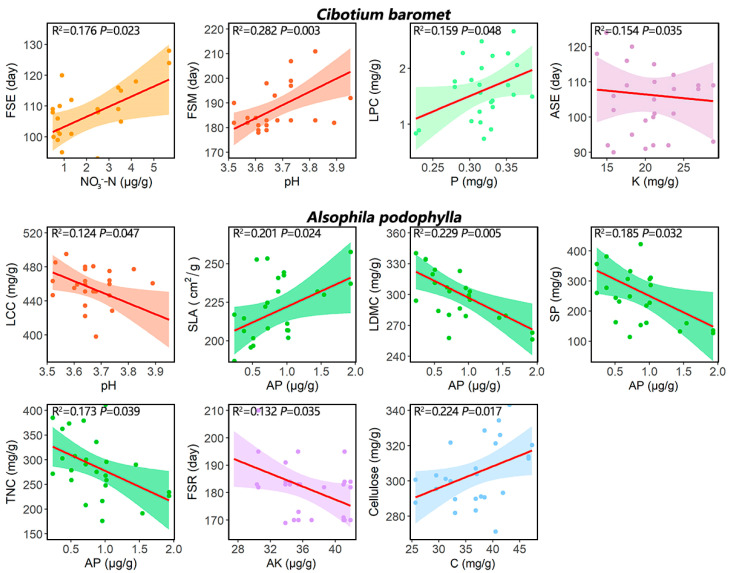
Relationships between the functional traits and their best micro-environment variable for the two REFs. Only the significant relationships (*p* < 0.05) were plotted. Abbreviations of the trait and ecological variables are, respectively, explained in Tables 3 and 4.

**Figure 3 plants-11-03320-f003:**
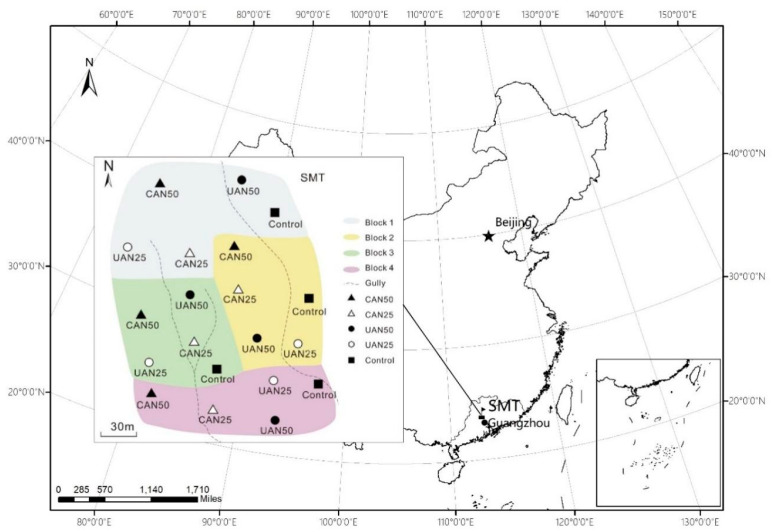
Location of the experimental site and plot map of the treatments. CAN25: canopy N addition at 25 kg N ha^−1^ yr^−1^, CAN50: canopy N addition at 50 kg N ha^−1^ yr^−1^, UAN25: understory N addition at 25 kg N ha^−1^ yr^−1^, UAN50: understory N addition at 50 kg N ha^−1^ yr^−1^.

**Table 1 plants-11-03320-t001:** Significance of multiple comparisons (*p* values) of N-addition rate and approach for each trait using Tukey HSD post hoc tests.

Category	Trait	*C. baromet*								*A. podophylla*							
		CAN			UAN			25	50	CAN			UAN			25	50
		0 vs. 25	0 vs. 50	25 vs. 50	0 vs. 25	0 vs. 50	25 vs. 50	CAN vs. UAN	CAN vs. UAN	0 vs. 25	0 vs. 50	25 vs. 50	0 vs. 25	0 vs. 50	25 vs. 50	CAN vs. UAN	CAN vs. UAN
Growth trait	H	0.82	<0.01 **	0.35	1.00	0.98	1.00	0.97	0.04 *	0.90	0.76	1.00	0.99	0.05 *	0.62	0.80	<0.01 ***
	SLA	0.89	1.00	0.81	1.00	0.58	0.57	0.89	0.47	0.65	0.99	0.41	1.00	0.97	0.99	0.55	1.00
	LCC	0.01 *	0.83	0.12	0.99	0.79	0.54	0.04 *	0.23	0.11	0.26	0.99	0.66	0.03 *	0.37	0.74	0.80
	LNC	0.99	0.92	0.70	0.93	0.63	0.97	0.72	0.98	1.00	0.33	0.34	0.99	0.48	0.76	0.99	1.00
	LPC	1.00	0.93	0.91	0.98	0.65	0.91	0.97	0.97	0.58	1.00	0.65	0.98	1.00	1.00	0.89	1.00
	Nitrate	0.45	0.04 *	0.67	0.32	0.35	1.00	1.00	0.77	0.01 *	0.86	0.06	1.00	0.33	0.22	<0.01 **	0.87
Defense trait	LDMC	1.00	0.99	1.00	1.00	0.62	0.44	0.98	0.85	0.47	0.98	0.20	0.83	1.00	0.81	0.09	0.97
	Lignin	0.98	0.74	0.42	0.99	1.00	0.94	1.00	0.90	<0.01 **	0.07	0.55	0.70	0.50	1.00	0.04 *	0.73
	Cellulose	0.61	0.01 *	0.25	<0.01 ***	0.07	0.15	0.01 *	0.94	0.26	0.99	0.46	0.90	0.92	1.00	0.74	0.99
	OA	0.31	0.04 *	0.82	0.24	0.03 *	0.79	1.00	1.00	0.03 *	1.00	0.03 *	0.60	1.00	0.65	<0.01 **	1.00
	SP	0.03 *	<0.01 **	0.86	0.38	<0.01 ***	0.01 *	0.65	0.66	0.61	0.98	0.29	1.00	0.94	0.92	0.58	0.67
	TNC	0.90	0.29	0.73	0.59	0.73	0.09	0.20	0.93	1.00	0.42	0.65	0.34	0.16	0.99	0.55	0.97
Reproductive trait	FSE	1.00	0.80	0.67	0.28	1.00	0.51	0.54	0.76	0.89	0.99	0.99	0.98	1.00	1.00	0.66	0.96
	ASE	0.49	1.00	0.54	1.00	1.00	1.00	0.54	1.00	1.00	0.99	0.98	1.00	0.05 *	0.05	1.00	0.26
	FSM	0.94	1.00	0.99	0.95	1.00	0.95	0.68	1.00	0.98	0.95	0.77	1.00	0.45	0.66	0.96	0.93
	ASM	0.54	1.00	0.51	0.99	0.97	0.89	0.84	1.00	0.98	0.67	0.93	1.00	0.04 *	0.11	1.00	0.64
	FSR	1.00	1.00	1.00	1.00	0.83	0.70	0.99	0.91	0.83	0.72	1.00	1.00	0.76	0.89	0.94	1.00
	ASR	0.99	<0.01 **	<0.01 **	0.03 *	0.47	0.86	0.04 *	0.05 *	0.93	1.00	1.00	0.99	0.73	0.56	0.79	0.94
	RD	0.99	0.12	0.35	<0.01 **	0.46	0.18	0.02 *	0.91	0.43	1.00	0.65	0.92	0.99	0.80	0.18	1.00

Abbreviations for the traits are defined in Table 3. CAN: canopy N addition, UAN: understory N addition; 0: 0 kg N ha^−1^ yr^−1^, 25: kg N ha^−1^ yr^−1^, 50: kg N ha^−1^ yr^−1^. “*”, “**” and “***” indicate statistical significance at *p* < 0.05, 0.01 and 0.001 respectively.

**Table 2 plants-11-03320-t002:** Models that best explained the traits of the studied REFs. Delta AICc values of the models are indicated.

Category	Traits	*C. baromet*					*A. podophylla*				
		N Treatment	Soil	Subcanopy Cover	Soil × N Treatment	Subcanopy Cover × N Treatment	N Treatment	Soil	Subcanopy Cover	Soil × N Treatment	Subcanopy Cover × N Treatment
Growth trait	H	1.31	**0.00**	2.90	9.37	11.16	**0.00**	1.67	14.83	4.57	1.56
	SLA	8.05	**0.00**	3.32	28.06	31.49	8.38	**0.00**	3.38	19.14	27.90
	LCC	**0.00**	3.48	6.94	11.33	12.95	3.15	**0.00**	4.66	3.80	25.43
	LNC	7.41	1.46	**0.00**	25.43	27.60	5.54	**0.00**	1.72	24.92	25.15
	LPC	10.08	**0.00**	2.33	19.45	28.46	9.86	**0.00**	3.58	19.6	24.5
	Nitrate	0.17	**0.00**	0.01	21.96	17.16	**0.00**	7.61	9.34	24.33	19.50
Defense trait	LDMC	9.69	**0.00**	3.96	24.66	28.89	4.04	**0.00**	3.16	23.20	23.49
	Lignin	5.26	**0.00**	0.33	25.42	28.79	**0.00**	6.56	8.02	19.28	20.56
	Cellulose	**0.00**	11.99	13.6	10.87	20.51	8.7	**0.00**	2.27	20.82	20.24
	OA	**0.00**	2.10	2.88	22.77	23.04	**0.00**	7.54	10.03	19.41	20.22
	SP	**0.00**	4.27	3.19	11.55	22.03	43.4	**0.00**	37.95	50.72	60.88
	TNC	**0.00**	0.93	1.31	22.35	24.17	6.11	**0.00**	4.97	20.99	25.04
Reproductive trait	FSE	6.24	**0.00**	2.93	19.48	20.82	7.44	**0.00**	2.30	20.38	22.05
	ASE	8.44	**0.00**	2.70	27.62	23.84	0.83	**0.00**	3.03	11.65	10.35
	FSM	17.03	**0.00**	9.53	17.25	33.82	**0.00**	6.78	3.77	16.51	12.89
	ASM	7.06	0.40	**0.00**	19.44	20.97	0.60	**0.00**	2.84	11.38	10.08
	FSR	10.07	**0.00**	2.93	23.91	26.33	9.69	**0.00**	4.61	16.76	15.60
	ASR	**0.00**	10.16	8.17	18.33	14.92	4.93	**0.00**	2.13	4.69	4.41
	RD	**0.00**	7.13	8.83	13.99	17.78	2.92	**0.00**	0.70	15.57	14.55

Abbreviations of traits are explained in Table 3. For each trait, only the selected best soil chemical variable (Appendix A) was applied to the models for “Soil” and “Soil × N treatment”. “Soil” is the best soil chemical variable selected among all the tested soil chemical variables (pH, C, N, P, K, NO_3_^−^-N, NH_4_^+^-N, AP, AK) for each trait (see Appendix A). The best models (with the smallest AICc values) are in bold.

**Table 3 plants-11-03320-t003:** Background information on the traits of the rare and endangered fern species (REFs) assessed in this study.

Category	Trait	Abbreviation	Specific Function
Growth traits	Plant height (cm)	H	Light capture and competition capacities [12,82]
	Specific leaf area (cm^2^/g)	SLA	Light capture and gaseous exchange capacities [12,77]
	Leaf carbon concentrations (mg/g)	LCC	Carbon fixation capacity [17,83]
	Leaf nitrogen concentrations (mg/g)	LNC	Maximum photosynthetic rate [12,77]
	Leaf phosphorus concentrations (mg/g)	LPC	Maximum photosynthetic rate [12,77]
	Leaf nitrate concentrations (mg/g)	Nitrate	Chlorophyll synthesis [12,84]
Defense traits	Leaf dry matter content (mg/g)	LDMC	Resistance to herbivory and drought [12,77,85]
	Leaf cellulose concentrations (mg/g)	Cellulose	Resistance to herbivory [26,86]
	Leaf lignin concentrations (mg/g)	Lignin	Resistance to herbivory [8,26]
	Leaf organic acid concentrations (mg/g)	OA	Resistance to oxidation by air pollutants [17,37]
	Leaf soluble phenolics concentrations (mg/g)	SP	Resistance to microbial pathogens and herbivory [8,37]
	Leaf total non-structural carbohydrates (mg/g)	TNC	Resistance to drought [14,52]
Reproductive traits	First spore emergence date (day)	FSE	Reproductive phenology, niche separation of phenology, and reproductive efficiency [42,56,80,81]
	All-spore emergence date (day)	ASE	
	First spore maturation date (day)	FSM	
	All-spore maturation date (day)	ASM	
	First spore release date (day)	FSR	
	All-spore release date (day)	ASR	
	Reproductive duration (day)	RD	

**Table 4 plants-11-03320-t004:** Ecological variables used in the models.

Category	Variable	Explanation
N treatment	N treatment	Canopy N addition at 25 and 50 kg N ha^−1^ yr^−1^ (CAN25 and CAN50), understory N addition at 25 and 50 kg N ha^−1^ yr^−1^ (UAN25 and UAN50), and a control
Micro-environment	Soil chemical property	Soil pH (pH), soil organic carbon content (C, mg/g), Soil total nitrogen content (N, mg/g), Soil total phosphorus content (P, mg/g), Soil total potassium content (P, mg/g), Soil nitrate nitrogen content (NH_4_^+^-N, μg/g), Soil ammonium nitrogen content (NO_3_^−^-N, μg/g), Soil available phosphorus content (AP, μg/g), Soil available potassium content (AK, μg/g)
	Subcanopy cover	Subcanopy cover (%)

## Data Availability

The data presented in this study are available on request from the corresponding author.

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
