# Peer review of "Effects of Simulated Nitrogen Deposition and Micro-Environment on the Functional Traits of Two Rare and Endangered Fern Species in a Subtropical Forest"

_plants, 2022, doi:10.3390/plants11233320_

Round 1

Reviewer 1 Report

Lines 69-70: "Because these methods of N addition bypass the interception of N by the forest canopy, they could induce incorrect assessments of the effects of N deposition on the forest ecosystem and soil chemical properties [7,27]." "incorrect" could be misleading here. Canopy N addition better simulation N deposition to forests,  but understory N addition has the merit of mimicing N entry into forest soils after N deposition intercepted by the canopy. you can say "Because these methods of N addition bypass the interception of N by the forest canopy, they could exagerate effects of N deposition on the forest ecosystem and soil chemical properties by adding all N directly to the soil without any reduction by the canopy".

Line 209: better give the P values of these compasion for each parameter before Figure 1. Similar for all other comparsions of significant or non-significant ones. 

Lines 215-216:  "Within each N addition rate, the effects of CAN on C. barometz traits were always greater than those of UAN." by "always" do you mean all parameters responded to CAN more than to UAN? in fact this is not always the case, if you compare response extent (whether increase or decrease compared to control), some changing percentage is larger by UAN than CAN (your lines 218-219). So I suggest to use "mostly" instead of  "always" to give a more precise stament of this trend. 

Lines224-225: again this is not always the case. rephrase it. 

Lines247-253: be careful about these statements which are too strong ones implying causal relationships. in fact these correlations could be inter-correlation relationships happening after N treatments. similar for lines 335-338, hardly can SLA cause changes in leaf defense traits. they are all leaf traits changing simultaneously, possibly driven by the same factor. 

Lines 269- : some comparsion and discussion on the differences on growth type (fast or slow growing species), nutrient demand type (conservative or high-demanding ones) between the two fern species would be necessary to be given at the beginning of the "Discussion" section, as well as in the "introduction". Such comparion highligh the significance of studying these two species (introduction) and may shed light on their different responses to N treatments.

Lines 321-324: it is rare that 8 years of continuous N input didn't cause any change soil properties. better give more reason to explain what's the mechanism behind. 

Lines 346-348: some discussion on the possible changes in understory plant composition or fern diversity would be necessary to add here based on your results, also add to the Abstract.

Author Response

Response to Reviewer 1 Comments

We thank the reviewer for the comments and suggestions. We have carefully considered all the comments and revised our manuscript accordingly. Revisions in our resubmitted manuscript have been highlighted.

Point 1: Lines 69-70: "Because these methods of N addition bypass the interception of N by the forest canopy, they could induce incorrect assessments of the effects of N deposition on the forest ecosystem and soil chemical properties [7,27]." "incorrect" could be misleading here. Canopy N addition better simulation N deposition to forests, but understory N addition has the merit of mimicing N entry into forest soils after N deposition intercepted by the canopy. you can say "Because these methods of N addition bypass the interception of N by the forest canopy, they could exagerate effects of N deposition on the forest ecosystem and soil chemical properties by adding all N directly to the soil without any reduction by the canopy".

Response 1: We appreciate for your valuable suggestion. The original “Because these methods of N addition bypass the interception of N by the forest canopy, they could induce incorrect assessments of the effects of N deposition on the forest ecosystem and soil chemical properties [7,27] ” has been changed to “Because these methods of N addition bypass the interception of N by the forest canopy, they could exaggerate effects of N deposition on the forest ecosystem and soil chemical properties by adding all N directly to the soil without any reduction by the canopy[9,29]” in page 2 line 70-73.

Point 2: Line 209: better give the P values of these compasion for each parameter before Figure 1. Similar for all other comparsions of significant or non-significant ones.

Response 2: Thank you for suggestion. To show the P values, we have moved the original Table S2 in supplementary material to main text as “Table 3. Significance of multiple comparisons (P values) of N addition rate and approach for each trait through Tukey HSD post hoc tests.” in page 7 line 256-260.

Point 3: Lines 215-216: "Within each N addition rate, the effects of CAN on C. barometz traits were always greater than those of UAN." by "always" do you mean all parameters responded to CAN more than to UAN? in fact this is not always the case, if you compare response extent (whether increase or decrease compared to control), some changing percentage is larger by UAN than CAN (your lines 218-219). So I suggest to use "mostly" instead of  "always" to give a more precise stament of this trend.”

Response 3: Thank you for pointing out this problem in manuscript, the "mostly" is more precise than "always". The original “Within each N addition rate, the effects of CAN on C. barometz traits were always greater than those of UAN.” has been changed to “Within each N addition rate, the effects of CAN on C. barometz growth traits were mostly greater than those of UAN” in page 6 line 230.

Point 4: Lines224-225: again this is not always the case. rephrase it.

Response 4: The original “Within each N addition rate, the effects of CAN on A. podophylla traits were always greater than those of UAN.” has been changed to “Within each N addition rate, the effects of CAN on A. podophylla traits were often greater than those of UAN.” in page 6 line 239-240.

Point 5: Lines247-253: be careful about these statements which are too strong ones implying causal relationships. in fact these correlations could be inter-correlation relationships happening after N treatments. similar for lines 335-338, hardly can SLA cause changes in leaf defense traits. they are all leaf traits changing simultaneously, possibly driven by the same factor.

Response 5: We agree and thank your kind suggestion. The original “Concerning the effects of dominant micro-environmental factors, for soil N effects (N, NO3--N and NH4+-N), we found that higher soil NO3--N delayed FSE for C. barometz (Figure 2). Lower soil pH advanced FSM for C. barometz and increased LCC for A. podophylla (Figure 2). For the other soil nutrient effects (C, P, AP, K and AK), C. barometz LPC decreased with decreasing soil P (Figure 2). The decreasing soil AP decreased A. podophylla SLA, but increased its TNC, LDMC and SP (Figure 2). Decreasing soil K delayed C. barometz ASE (Figure 2) and decreasing soil AK delayed A. podophylla FSR (Figure 2). However, A. podophylla Cellulose increased with decreasing soil C (Figure 2). Subcanopy cover had no effects on the traits of either REF (Table S4-6)” has been changed to “Concerning the effects of micro-environmental factors (Figure 3 and Table S4-S6 in Supplementary Material), for soil N effects (N, NO3--N and NH4+-N), we found that higher soil NO3--N can be related to delayed FSE for C. barometz. Lower soil pH could advance FSM for C. barometz and increase LCC for A. podophylla. For the other soil nutrient effects (C, P, AP, K and AK), C. barometz LPC decreased with decreasing soil P. The decreased A. podophylla SLA, but its increased TNC, LDMC and SP were found to increase with soil AP. Decreasing soil K showed the tendency to delay C. barometz ASE, and decreasing soil AK was related to delayed A. podophylla FSR. However, A. podophylla Cellulose increased with decreasing soil C. Subcanopy cover had no effects on the traits of either REF.” in page 8-9 line 283-291.

The original “However, contrary to our hypothesis, lower soil pH advanced C. barometz FSM and increased A. podophylla LCC, possibly because both REFs prefer more acidic soil than that indicated by a pH value in the normal range [36,78]” has been changed to “However, lower soil pH could advance C. barometz FSM and increase A. podophylla LCC, possibly because both REFs prefer more acidic soil than that indicated by a pH value in the normal range [40,84]” in page 12 line 393-395.

Point 6: Lines 269- : some comparsion and discussion on the differences on growth type (fast or slow growing species), nutrient demand type (conservative or high-demanding ones) between the two fern species would be necessary to be given at the beginning of the "Discussion" section, as well as in the "introduction". Such comparion highligh the significance of studying these two species (introduction) and may shed light on their different responses to N treatments.

Response 6: We thank your helpful suggestion. We had already written “The two REFs are both fern species associated to subtropical areas [40], with A. podophylla has a funnel-shaped crown and aerial roots while C. barometz does not.” in the original introduction in page 2 line 87-89.

We have also added The two REFs we studied are fern species widely distributed in subtropical region [40]. Yet, there were some differences in their morphological and physiological structure. For example, A. podophylla has a funnel-shaped crown and aerial roots while C. barometz does not.” to the beginning of discussion in page 10 line 305-308.

Point 7: Lines 321-324: it is rare that 8 years of continuous N input didn't cause any change soil properties. better give more reason to explain what's the mechanism behind.

Response 7: Thank you for your comment. The result was counter-intuitive and we have added the explanation. The original “Contrary to our hypothesis, N addition did not result in any significant changes in micro-environment factors at our experimental site after the 8-year N addition treatment. This indicates that N deposition did not immediately induce any strong variations in the micro-environment. This was also found to be the case in other forest ecosystems like temperate deciduous forests” has been changed to “Contrary to our hypothesis, N addition did not result in any significant changes in micro-environment factors at our experimental site after the 8-year N addition treatment. This can be explained by a very recent study at our experimental plots by Tian et al. [77]. They found that plants especially tree species absorbed and stored substantially additional N from soil by their roots (increased the plant N pool by 120–412 %), so that soil N did not show significant variation under N addition treatment. The insignificant changes of soil properties in response to N addition could also be found in other forest ecosystems. For example, a study by Yang et al [78] in temperate forests showed that the 9-year N addition (100 kg N ha-1 yr-1) did not cause changes in soil N and pH, and they also attributed this to plants’ absorption of additional N. In addition, some studies (in subtropical or temperate forests) showed that only under very high N addition rates (100 and 150 kg N ha-1 yr-1 in their studies vs 50 kg N ha-1 yr-1 of ours) can soil pH and N be affected [4,10,23].” in page 11-12 line 371-382.

Point 8: Lines 346-348: some discussion on the possible changes in understory plant composition or fern diversity would be necessary to add here based on your results, also add to the Abstract.

Response 8: We gratefully appreciate for your valuable suggestion. We have added “this may cause the increasing A. podophylla population and decreasing C. baromet population in subtropical forests.” to abstract in page 1 line 29-30.

We have also added “These results implied that the population of A. podophylla may increase and that of C. barometz face the possibility of declining under high N deposition in subtropical forests.” to conclusion in page 12 line 403-405.

Reviewer 2 Report

General

The paper titled “Effects of simulated nitrogen deposition and micro-environment on the functional traits of two rare and endangered fern species in subtropical forest” reported about the connections between the growth, defence and reproductive strategies of two fern species and nitrogen treatments in a forest. These studies are important as the problem of nitrogen deposition is very threatening in our modern world. The manuscript is well-written, the study raises good questions and investigates relevant variables. However, I have some questions and comments and some suggestions related to the statistical analyses.

Details

Abstract:

Line 24: o on?

Introduction

Lines 35-37: reference!

Lines 39-40: knowledge related to the effects of N deposition?

Lines 42-43: Do the approaches reflect the strategies, or do the traits reflect the strategies?

Line 51: impact of effect?

Lines: 71-75: What did these studies find? Is N deposition harmful or favourable?

Lines 78-106: It is too long and too much! Shorten it!

Methods:

Lines 113-115: descriptors of the species such as: Castanopsis eyrei (Champ. ex Benth.)

Lines 129-130: What was the reason for blocks? Blocks were not taken into models and some plots were closer to each other between two blocks then within a block. What were the aspects to choose the plots?

Line 129: I suggest to take Figure S1 into the manuscript.

Lines 131-132: Why didn’t you apply the two types of treatments (canopy and understorey) together?

Line 134: You should briefly describe the spraying methods here.

Line 140. “After surveying the stand” What does it mean?

Line 156: Were these 3-5 individuals of ferns representative for one plot? I am really interested in the density of these ferns in a plot? 15 leaves from one individual or all together 15 leaves from 3-5 individuals? Were the same leaves used for all the analyses or for defence traits new leaves were collected?

Line 173: again the question: were these three individuals representative for the plots?

Table S4: first row: P and pH?

Statistical analyses:

The Authors worked with many variables and I don’t think that they always used the proper methods to analyse them. ANOVAs were okay but it is not a good idea to analyse all response variables separately and independently with all explanatory variables. I suggest to see at first the connections between the traits, whether there are some correlations between them, and then also correlations among the micro-environmental variables. Don’t use just AIC values, whether significant the model or not. If there is no significant connections between the variables, then the delta AIC is also irrelevant. And what does it mean “best” variable? “Best” model? Best for you? I suggest to perform e.g. ANCOVAs for each traits but I can also imagine multivariate analyses for all the traits and descriptive variables. Think of models with several variables, not everything separately!

Results

Lines 207, 209, 213: do not list again these traits…

Lines 215, 225: “always greater”: “always” is an excess because not all traits had responses.

Figure 1: C. baromet: Nitrate plot: how come two “a” for the very different UAN0 and UAN25?

ASR plot: how come two “A” for the very different CAN25 and UAN25. Recheck them!

Line 247: what is a dominant micro-environmental variable?

Lines 235-255: Why is it so that there was no connection between the nitrogen treatment and the micro-environmental variables? Then what do we expect?

Discussion

I suggest not to use all abbreviations of traits in the section Discussion. It would be easier to read their names finally.

Lines 274-275: and nitrate: UAN25?

Line 282: wasn’t this your result? Why are there these references?

Lines 286-290: Why is this difference between the results of canopy and understorey?

Line 302: Was A podophylla under stress because of the limited nitrogen?

Lines 304-306: Why does A podophylla “want to” improve its reproductive traits as it has better circumstances with higher nitrogen content?

Lines 310-311: explain? It is just an assumption or did you investigate it?

Lines 312-319: Are there any connections between the canopy layers and the traits?

Line 320: REF traits

Line 323: It often happens that adding nitrogen or other fertilizers has instant (within 1-2 years) effects on plants or animals but in long-term this effect cannot be detected any more.

Lines 327-329: In context of or irrespective of N deposition? There was no connections between N deposition and micro-environmental variables.

Lines 351-354: Microenvironment’s influence can override the effects of N addition? Then what can the N addition directly affect?

Author Response

Response to Reviewer 2 Comments

The paper titled “Effects of simulated nitrogen deposition and micro-environment on the functional traits of two rare and endangered fern species in subtropical forest” reported about the connections between the growth, defence and reproductive strategies of two fern species and nitrogen treatments in a forest. These studies are important as the problem of nitrogen deposition is very threatening in our modern world. The manuscript is well-written, the study raises good questions and investigates relevant variables. However, I have some questions and comments and some suggestions related to the statistical analyses.

Response: We thank the reviewer for the comments and suggestions. We have carefully considered all the comments and revised our manuscript accordingly. Revisions in our resubmitted manuscript have been highlighted.

Details

Abstract:

Point 1: Line 24: o on?

Response 1: Corrected.

Introduction:

Point 2: Lines 35-37: reference!

Response 2: We have cited references [1,2] in page 1 line 38, and have added them to reference list to page 13, line 442-446.

  1. Battye, W.; Aneja, V.P.; Schlesinger, W.H. Is nitrogen the next carbon? 2017, 5, 894-904, doi:https://doi.org/10.1002/2017EF000592.
  2. Steffen, W.; Richardson, K.; Rockstrom, J.; Cornell, S.E.; Fetzer, I.; Bennett, E.M.; Biggs, R.; Carpenter, S.R.; de Vries, W.; de Wit, C.A.; et al. Planetary boundaries: Guiding human development on a changing planet. Science 2015, 347, doi:10.1126/science.1259855.

Point 3: Lines 39-40: knowledge related to the effects of N deposition?

Response 3: Thank you for your comment. The original “Knowledge is lacking for rare and endangered fern species (REFs).” has been changed to “Knowledge is lacking related to the effects of N deposition on rare and endangered fern species (REFs).”in page 1 line 40-41.

Point 4: Lines 42-43: Do the approaches reflect the strategies, or do the traits reflect the strategies?

Response 4: Thank you for your comment. What we want to express here is that traits reflect the strategies. The original “When studying the effects of environmental change, including N deposition, on forest plant species, trait-based approaches are increasingly being used since they reflect plant growth, defense and reproductive strategies [9].” has been changed to “When studying the effects of environmental change, including N deposition, on forest plant species, trait-based approaches are increasingly being used since traits reflect plant growth, defense and reproductive strategies [11].” in page 1-2 line 43-45.

Point 5: Line 51: impact of effect?

Response 5: The original “However, studies to date have concentrated mainly on the responses of dominant species, with research gaps on the impact of the N deposition effect on the functional traits of REFs [1,3,15,16].” has been corrected as “However, studies to date have concentrated mainly on the responses of dominant species, with research gaps on the impact of the N deposition on the functional traits of REFs [3,5,17,18].”in page 2 line 51-53.

Point 6: Lines: 71-75: What did these studies find? Is N deposition harmful or favourable?

Response 6: Thank you for your comment. We have added “Among those studies, canopy N addition can induce increased survival stresses for some understory plants [5,17]; it could also promote the diversity of soil biota and macrofauna but have no impact on plant diversity [15,31]. Canopy N addition promoted or had no impact on the growth of dominant canopy tree species growth, depending on specific tree species [18,38]. Moreover, compared with CAN, UAN always had greater effects on dominant understory species, but had weaker effects on dominant tree species and soil fauna [17,18,38,39].” to page 2 line 77-83.

Point 7: Lines 78-106: It is too long and too much! Shorten it!

Response 7: We agree and thank your suggestion, and we have shortened it as below in page 2-3 line 85-103:

The objective of this study was to investigate the impacts of simulated N deposition on the growth, defense and reproductive traits (19 traits in total) of two REFs (Cibotium barometz and Alsophila podophylla). The two REFs are both fern species associated to subtropical areas [40], with A. podophylla has a funnel-shaped crown and aerial roots while C. barometz does not. We also compared the ability of N addition and micro-environmental conditions (canopy cover and soil properties) to explain the variation in the REFs’ functional traits. Our experiment was conducted in an evergreen broad-leaved forest in subtropical South China, where the CAN (canopy N addition) and UAN (understory N addition) were applied.  

We attempted to test the following hypotheses:

1) N addition would decrease the growth traits of both REFs, and increase their defense traits due to the environmental stress induced by N addition. For reproductive traits, N addition would delay reproductive events and prolong reproductive duration [3,41]. Meanwhile, under the same N addition rate, the effect on traits would be greater for UAN than for CAN.

2) N addition would directly increase soil nitrogen contents but reduce soil pH and other soil nutrients due to N’s potential effect on soil acidification and to soil nutrient leaching or imbalance. Accordingly, micro-environmental conditions would also have significant impact on the functional traits of both REFs [17,42,43].

Methods:

Point 8: Lines 113-115: descriptors of the species such as: Castanopsis eyrei (Champ. ex Benth.)

Response 8: Thank you for your suggestion. We have added descriptors of the species. The original “The two REFs - Cibotium barometz and Alsophila podophylla; The dominant tree species are Castanea henryi, Castanopsis eyrei and Schima superba; dominant shrubs include Blastus cochinchinensis, Psychotria rubra and Ardisia quinquegona; and the dominant herbaceous species are Alpinia chinensis, Hypolytrum latifolium and Sarcandra glabra.” has been changed to “The two REFs - Cibotium barometz (L.) J. Sm. and Alsophila podophylla Hook.; The dominant tree species are Castanea henryi (Skan) Rehd. et Wils., Castanopsis eyrei (Champ. ex Benth.) Tutch. and Schima superba Gardn. et Champ.; dominant shrubs include Blastus cochinchinensis Lour. Psychotria rubra Wall. and Ardisia quinquegona Blume; and the dominant herbaceous species are Alpinia chinensis Hayata, Hypolytrum latifolium L. C. Rich. and Sarcandra glabra (Thunb.) Nakai.” in page 3 line 110-114.

Point 9: Lines 129-130: What was the reason for blocks? Blocks were not taken into models and some plots were closer to each other between two blocks then within a block. What were the aspects to choose the plots?

Response 9: Thank you for your comment. We have added “Four blocks were established at each forest site, and each treatment was replicated once within each of the four blocks. Within each block, the five treated plots were randomly assigned.” to experimental design in page 3 line 128-130.

Furthermore, according to your suggestion, we did ANOVAs with block effect considered (Table S1) to compare the effects of the N addition treatments on each trait and two micro-environmental variables (soil chemical properties and subcanopy cover). The block effects were not significant, and adding block effect did not change the results of variation in traits or micro-environment to treatments (see Table S1 below as an example).

We have revised the description of the analysis method. Specifically, the original “Analyses of variance (ANOVAs)” has been changed to “Analyses of variance (ANOVAs) considering block effect” to page 5 line 202.

We have also added “The block effects were not significant on the traits of both C. barometz and A. podophyll” to Results in page 6 line 242-244.

“Table S1. Analyses of variance showing treatment and block effects on traits of the two REFs” has been added to Supplementary Material in page 6.

Table S1. Analyses of variance showing treatment and block effects on traits of the two REFs

Category

Trait

Effect variable

C. baromet

A. podophylla

Mean Sq

P

Mean Sq

P

Growth trait

H

N treatment

16228.00

<0.01

33640.08

<0.01

Block

1092.88

0.84

271.07

0.98

SLA

N treatment

484.32

0.39

643.86

0.40

Block

534.74

0.34

60.38

0.96

LCC

N treatment

1203.88

<0.01

2048.65

0.03

Block

278.16

0.24

1071.14

0.19

LNC

N treatment

3.89

0.40

7.81

0.25

Block

3.19

0.47

1.30

0.86

LPC

N treatment

0.22

0.59

0.26

0.54

Block

0.31

0.40

0.63

0.16

Nitrate

N treatment

0.91

0.05

2.01

0.01

Block

0.80

0.08

0.11

0.83

Defense trait

LDMC

N treatment

358.54

0.55

1383.86

0.14

Block

210.70

0.71

430.77

0.61

Lignin

N treatment

2.45

0.49

25.03

<0.01

Block

0.57

0.89

6.06

0.27

Cellulose

N treatment

12.06

<0.01

3.90

0.28

Block

0.85

0.68

5.18

0.17

OA

N treatment

66.13

0.02

169.86

<0.01

Block

38.02

0.11

18.54

0.62

SP

N treatment

24736.38

<0.01

16591.12

0.25

Block

3597.83

0.27

24198.06

0.13

TNC

N treatment

4614.57

0.03

6995.11

0.05

Block

886.95

0.59

8501.21

0.10

Reproductive trait

FSE

N treatment

274.05

0.10

142.76

0.67

Block

256.74

0.14

263.53

0.36

ASE

N treatment

155.79

0.44

1079.08

0.04

Block

100.46

0.60

200.05

0.65

FSM

N treatment

61.31

0.66

345.76

0.31

Block

365.76

0.10

306.27

0.36

ASM

N treatment

689.32

0.30

683.06

0.06

Block

1083.96

0.14

128.82

0.69

FSR

N treatment

59.21

0.76

121.61

0.58

Block

142.16

0.36

219.83

0.29

ASR

N treatment

836.24

<0.01

287.94

0.58

Block

142.18

0.26

340.52

0.47

RD

N treatment

867.12

0.01

265.57

0.29

Block

66.28

0.77

32.32

0.92

Abbreviations for the traits are defined in Table 1. Bold values indicate statistical significance at P < 0.05.

Point 10: Line 129: I suggest to take Figure S1 into the manuscript.

Response 10: Thank you for your suggestion. We have added Figure S1 into the manuscript in page 4, line 146-149.

Point 11: Lines 131-132: Why didn’t you apply the two types of treatments (canopy and understorey) together?

Response 11: Thank you for your constructive suggestion. The original intention of our N addition experimental platform was to design a novel method with canopy vs. understory N treatment and detect the merits and pitfalls of the two approaches. This design was based on the fact that very few researchers attempt to simulate the natural N deposition processes in forest canopies. Your suggestion on applying both canopy and understory treatments is very interesting and could be a novel and advanced experimental design, which we can consider in future study.

Point 12: Line 134: You should briefly describe the spraying methods here.

Response 12: Thank you for your comment. We have added description of spraying methods “In brief, the CAN was applied via a 35-m high tower (5–8 m above the canopy, with 4 sprinklers) built in the center of each CAN plot, while the UAN was applied by five sprinklers that were evenly distributed 1.5 m above the ground in each UAN plot. These sprinklers could turn 180-360° and spray the N solution as far as 17 m, depending upon the pressure used. The targeted N solution was made by weighing appropriate amount of ammonium nitrate (NH4NO3) and mixed with surface water drained from a nearby pond, and was transported to sprinklers by various polyvinylchloride pipes.” to page 3 line 135-142.

Point 13: Line 140. “After surveying the stand” What does it mean?

Response 13: Thank you for your comment. The “stand” is a more forestry term and we changed it to “vegetation”. The original “After surveying the stand in each quadrat and classifying each stand into two strata” has been changed to “After surveying the vegetation in each quadrat and classifying it into two strata” in page 4 line 152.

Point 14: Line 156: Were these 3-5 individuals of ferns representative for one plot? I am really interested in the density of these ferns in a plot? 15 leaves from one individual or all together 15 leaves from 3-5 individuals? Were the same leaves used for all the analyses or for defence traits new leaves were collected?

Point 15: Line 173: again the question: were these three individuals representative for the plots?

Response 14&15: The two REFs were the understory dominant species in our studied forest. There are on average 4-5 mature individuals of C. baromet or A. podophylla in each 10m × 10m plot. The two REFs often have large crown of which the diameter can reach about 2 m. The 3-5 individuals of ferns are representative for one plot based on their occurrence and cover. To make it clear, the original “We then collected 15 leaves from 3-5 individuals per species per quadrat.” has been changed to “We then selected 3-5 mature individuals per species per quadrat, and collected 15 leaves from each individual. The same leave samples were used to analyses growth traits and defense traits.” in page 4 line 168-170.

Point 16: Table S4: first row: P and pH?

Response 16: The original “pH” has been corrected as “P” in Table S4 in Supplementary material in page 8.

Statistical analyses:

Point 17: The Authors worked with many variables and I don’t think that they always used the proper methods to analyse them. ANOVAs were okay but it is not a good idea to analyse all response variables separately and independently with all explanatory variables. I suggest to see at first the connections between the traits, whether there are some correlations between them, and then also correlations among the micro-environmental variables. Don’t use just AIC values, whether significant the model or not. If there is no significant connections between the variables, then the delta AIC is also irrelevant. And what does it mean “best” variable? “Best” model? Best for you? I suggest to perform e.g. ANCOVAs for each traits but I can also imagine multivariate analyses for all the traits and descriptive variables. Think of models with several variables, not everything separately!

Response 17: We gratefully appreciate for your comments on our data analysis suggestion. We agreed that the correlations between traits or soil properties are also interesting to be shown. Therefore, we have added “Figure S2 Correlations between traits of C. baromet”, “Figure S3. Correlations between traits of A. podophylla”, and “Figure S4. Correlations between soil chemical properties” to page 2-4 in Supplementary material.

We have also added the text about the results of correlation analysis of traits as below:

We have added “The correlations between traits and between soil chemical properties were determined through Pearson correlation analysis.” to statistical analyses in page 5, line 207-208.

The “For the correlations between traits, the three growth traits of C. baromet - SLA, LNC and LPC were positively correlated between each other, and each of them also showed negative correlation with the defense trait of LDMC (Figure S2 in Supplementary Material). Meanwhile, negative relationships between growth traits and defense traits or reporductive traits could also be found. In contrast, there was no correlation between A. podophylla growth traits, but negative correlations between its defense traits and postive relations between its reproductive traits were often found (Figure S3 in Supplementary Material). Meanwhile, growth traits often have significant relationships with defense traits, with both positive or negative relationships could be detected depending on the specific traits.” has been added to results in page 6-7 line 245-254.

The ‘’Positive correlations were often found between soil properties (Figure S4 in Supplementary Material). Yet, soil pH frequently showed negative correlations with the other soil properties (except for its strongly positive relationship with NO3--N). Both soil C and N positively correlated with soil P, AP, NH4+-N and AK.” has been added to results in page 8 line 270-273.

For your questions related to AIC values and multivariate analyses, we appreciate your very insightful suggestions. However, we have other considerations that we think it could also be reasonable to keep some parts of our previous analysis as it was. We choose to still keep AIC analysis because we would like to test one of our hypotheses on the relative importance of N deposition and Micro-environment. The role of micro-environment in explaining the REFs’ traits variations under N addition treatment was not well understood. Meanwhile, we had considered multivariate analyses in the very early stage of our data analysis, but we finally prefer to show results as more basic descriptions related to the specific effects of soil properties on each trait of the two REFs. From our perspective, until now, basic knowledge of those two important REFs is lacking, and we want to uncover more detailed information from traits perspective. This would also be helpful for further greenhouse-controlled experiments to select the proper levels of different interested soil variables and trait indices. In addition, since only those two species were REFs in our study area, we tried to give as detailed information as possible while answering our research question. In case that we have a lot of REFs in the same study area and would like to subtract key information for conserving all of them, we will use the method as you kindly suggested.

Figure S2. Correlations between traits of C. baromet. Abbreviations for the traits are defined in Table 1. ”*”, “**” and “***” indicate statistical significance at P < 0.05, 0.01 and 0.001 respectively.

Figure S3. Correlations between traits of A. podophylla. Abbreviations for the traits are defined in Table 1. ”*”, “**” and “***” indicate statistical significance at P < 0.05, 0.01 and 0.001 respectively.

Figure S4. Correlations between soil chemical properties. Abbreviations for the ecological variables are defined in Table 1. ”*”, “**” and “***” indicate statistical significance at P < 0.05, 0.01 and 0.001 respectively.

Results:

Point 18: Lines 207, 209, 213: do not list again these traits…

Response 18: Thank you for your comment. We have removed those lists of traits. The original “(H, SLA, LCC, LNC, LPC and Nitrate), (LDMC, Lignin, Cellulose, OA, SP and TNC) and (FSE, ASE, FSM, ASM, FSR, ASR and RD)” have been removed.

Point 19: Lines 215, 225: “always greater”: “always” is an excess because not all traits had responses.

Response 19: Thank you for your comments. The original “Within each N addition rate, the effects of CAN on C. barometz traits were always greater than those of UAN.” has been changed to “Within each N addition rate, the effects of CAN on C. barometz growth traits were mostly greater than those of UAN” in page 6 line 229-230.

The original “Within each N addition rate, the effects of CAN on A. podophylla traits were always greater than those of UAN.” has been changed to “Within each N addition rate, the effects of CAN on A. podophylla traits were often greater than those of UAN.” in page 6 line 239-240.

Point 20: Figure 1: C. baromet: Nitrate plot: how come two “a” for the very different UAN50 and UAN25?

Point 21: ASR plot: how come two “A” for the very different CAN25 and UAN25. Recheck them!

Response 20&21: We checked our data again, and found an outlier of ASR value for C. baromet. We removed this value and did the ANOVA again, and showed significant differences between CAN25 and UAN25 and between Control and UAN25, and we have corrected the original “A” as “B” and the original “a” as “b” in Figure 2. For Nitrate it was still not statistically significant between UAN50 and UAN25 after careful checking. The original (left) and corrected (right) plots of C. barometz ASR were as below:

We have also corrected relevant text about ASR in Results as below:

The original “ASR occurred significantly later under CAN50” has been changed to “ASR occurred significantly later under UAN25 and CAN50”in page 6 line 228-229.

The original “the effects of CAN on C. barometz traits were always greater than those of UAN.” has been changed to “the effects of CAN on the growth traits of C. barometz were greater than those of UAN” in page 6 line 230.

The original “There were two exceptions: the effects of UAN25 on Cellulose and RD were greater than those of CAN25 (Figure 1).” has been changed to “but the effect of CAN50 on ASR was greater than that of UAN50 (Figure 2).”in page 6 line 233-234.

Point 22: Line 247: what is a dominant micro-environmental variable?

Response 22: Thank you for your comment. The “dominant” has been removed.

Point 23: Lines 235-255: Why is it so that there was no connection between the nitrogen treatment and the micro-environmental variables? Then what do we expect?

Response 23: Thank you for your comment. The result was counter-intuitive and we have added the explanation as “Contrary to our hypothesis, N addition did not result in any significant changes in micro-environment factors at our experimental site after the 8-year N addition treatment. This can be explained by a very recent study at our experimental plots by Tian et al. [77]. They found that plants especially tree species absorbed and stored substantially additional N from soil by their roots (increased the plant N pool by 120–412 %), so that soil N did not show significant variation under N addition treatment. The insignificant changes of soil properties in response to N addition could also be found in other forest ecosystems. For example, a study by Yang et al. [78] in temperate forests showed that the 9-year N addition (100 kg N ha-1 yr-1) did not cause changes in soil N and pH, and they also attributed this to plants’ absorption of additional N. In addition, some studies (in subtropical or temperate forests) showed that only under very high N addition rates (100 and 150 kg N ha-1 yr-1 in their studies vs 50 kg N ha-1 yr-1 of ours) can soil pH and N be affected [4,10,23].” to discussion in page 11-12 line 371-382.

Discussion

Point 24: I suggest not to use all abbreviations of traits in the section Discussion. It would be easier to read their names finally.

Response 24: Thank you for your suggestion. We have used the full name when traits were first mentioned, and then used their abbreviations. The original “H, LCC and Nitrate” has been changed to “plant height, leaf carbon and nitrate concentrations (H, LCC and Nitrate)” in page 10 line 310-311.

The original “OA and Cellulose” has been changed to “leaf organic and cellulose concentrations (OA and Cellulose)” in page 10 line 321-322.

The original “CAN50 significantly delayed ASR and UAN25 significantly prolonged RD” has been changed to “N addition treatment (CAN50 or UAN25) significantly delayed all spore release date (ASR) and significantly prolonged reproductive duration (RD)” to avoid misunderstanding” in page 10-11 line 327-329.

The original “Lignin” has been changed to “leaf lignin concentration (Lignin)” in page 11 line 341.

The original “ASE and ASR” has been changed to “all spore emergence date and maturation date (ASE and ASR)” in page 11 line 344-345.

Point 25: Lines 274-275: and nitrate: UAN25?

Response 25: Thank you for your kind comment. We have rechecked nitrate in Figure 2, it was still only affected by CAN50.

Point 26: Line 282: wasn’t this your result? Why are there these references?

Response 26: Yes, it was our results, and we did not cite these references at the right place. We have moved them to the end of the sentence “This indicates that the applied N solution created a series of stresses (e.g., oxidative stress) on the C. barometz leaves [7,14,37].” in page 10 line 322-324.

Point 27: Lines 286-290: Why is this difference between the results of canopy and understorey?

Response 27: Thank you for your comment. The original “Concerning C. barometz reproductive traits, CAN50 significantly delayed ASR and UAN25 significantly prolonged RD” has been rephrased as “Concerning C. barometz reproductive traits, N addition treatment (CAN50 or UAN25) significantly delayed all spore release date (ASR) and significantly prolonged reproductive duration (RD)” to avoid misunderstanding in page 10-11, line 326-329.

Point 28: Line 302: Was A podophylla under stress because of the limited nitrogen?

Response 28: From our point of view, we are not sure if A podophylla was under stress because of the limited nitrogen, since we did not find any literature relevant to this topic. We just can infer based on our results, that more N than the average level of N in this study plots would promote the growth of A podophylla and might alleviate its environmental stress. For instance, previous studies found that N addition (100 kg ha-1 yr-1) can help some fern species invest more N to enhance P acquisition and mitigate P limitation (Li et al., 2012; Tian et al., 2018). This may also happen to A podophylla. This question also makes us think it would be a good idea that in future we could compare the N level of A podophylla habitat in other natural forests in subtropical area, to really know if this species was in fact in limited nitrogen condition, or to find the threshold of N level that is good for this species.

Point 29: Lines 304-306: Why does A podophylla “want to” improve its reproductive traits as it has better circumstances with higher nitrogen content?

Response 29: Thank you for your comment. Our inference was based on previous studies showed that, the increased resource availability by N addition can make plants allocate more resources (e.g., nutrients) to their reproductive organs, thus accelerate the development of reproductive organs and advance the reproductive phenology (Yang et al., 2020; Li et al., 2021). We have added “This was consistent with some previous studies on grasslands, which showed that the increased resource availability by N addition can make plants allocate more resources (e.g., nutrients) to their reproductive organs, thus accelerate the development of reproductive organs and advance the reproductive events [3,41,72,73].” to page 11 line 346-350.

Point 30: Lines 310-311: explain? It is just an assumption or did you investigate it?

Response 30: Thank you for your comment. It was what we investigated in our experimental plots: “the funnel-shaped crown and aerial roots we observed for A. podophylla, which are lacking for C. barometz”.

The original “Furthermore, the funnel-shaped crown and aerial roots we observed for A. podophylla, which are lacking for C. barometz, which also explain the different responses of their traits to nitrogen deposition.” has been supplemented as “Furthermore, the funnel-shaped crown and aerial roots were observed for A. podophylla, which were lacking for C. barometz. Those differences would reflect the varied ability of the two REFs to absorb or compete for natural resources (e.g., water and nutrients), hence resulting in their contrary responses to N addition [50,75,76]. Some studies on grassland also found that plant species with distinct resource acquisition ability (e.g., grass vs rare forbs) had contrary responses to N addition treatment [41,72,73].” to page 11 line 355-361.

Point 31: Lines 312-319: Are there any connections between the canopy layers and the traits?

Response 31: Yes, previous studies have reported the effects of canopy layer or canopy tree species on understory plant traits. For example, Jiang et al. (2022) found a positive correlation between canopy openness and the leaf phosphorus content of understory plants. Oguchi et al. (2017) found that the increase of canopy openness can improve the leaf light saturation rate of six deciduous woody species seedlings in temperate deciduous forest. Wang et al (2015) found that canopy species Cunninghamia lanceolata decreased diameters of 1- and 2-order fine roots of understory species Loropetalum chinensis by the integrated effects of nutrient competition and litter input. Furthermore, Forrister et al (2019) reported canopy species can affect the defense traits (leaf defense compound concentrations) of understory plants by sharing host-specific pests (e.g. herbivores and pathogens).

Point 32: Line 320: REF traits

Response 32: Thank you for your comment. The original “REFtraits” has been changed to “REF traits” in page 11 line 370.

Point 33: Line 323: It often happens that adding nitrogen or other fertilizers has instant (within 1-2 years) effects on plants or animals but in long-term this effect cannot be detected any more.

Response 33: We agree with your comment. Therefore, it could be the opposite situation to our inference. We therefore removed the original sentence “This indicates that N deposition did not immediately induce any strong variations in the micro-environment.” We have also added a more detailed explanation on the non-significant variations of soil properties among treatments (see response to Point 23).

Point 34: Lines 327-329: In context of or irrespective of N deposition? There was no connections between N deposition and micro-environmental variables.

Response 34: Thank you for your comment. We have removed the “in a context with N deposition” in this sentence. The original “Therefore, in a context with N deposition, the micro-environment needs to be considered in order to improve REF growth and reproduction in subtropical forests.” has been changed to “Therefore, the micro-environment needs to be considered in order to improve REF growth and reproduction in subtropical forests.” in page 12 line 385-386.

Point 35: Lines 351-354: Microenvironment’s influence can override the effects of N addition? Then what can the N addition directly affect?

Response 35: Thank you for your comment. The original “since its influence can override the effects of N addition” has been changed to “since its influence sometimes overrides the effects of N addition.” page 12 line 412-413.

Round 2

Reviewer 2 Report

I am satisfied with most of the responses of the Authors. They are very precise and accurate. They considered all my questions and suggestions. Most of the answers are good and I accept most of them. However, I have some comments following the Authors’ points.

Point 7: line 87: the bracket is not finished.

Point 9: Unfortunately, the Authors misunderstood my comments about blocks. I didn’t want to see the effects of blocks. On the contrary, I saw that blocks did not have any effects, and that is why I asked “what was the reason for blocks?” However, I’ve got the answer to my question about the aspects of choosing plots. I think that the ANOVAs are unnecessary here. They can be left with the new Table S1.

Point 15: I would add the information about the density of these plants to the manuscript.

Point 17:

Correlations: It is very nice that the Authors performed these correlations but did they find anything interesting or something worth to interpret about it? If the Authors do not want to add any interpretations about these correlations, then it makes no sense to keep them in the manuscript.

Delta AIC values: I am not convinced about the relevance of calculating delta AIC. I think this method is only for covering the absence of significant results, however, it is not a problem having insignificant results. If the Editor accepts this method, at least I suggest the Authors to write a short justification about this method into the Statistical analyses section or writing a reference where this method has already been used elsewhere.

Point 31: It is nice, but some of these interpretations could be added to the manuscript, as well.